# Extraction of Inulin from Andean Plants: An Approach to Non-Traditional Crops of Ecuador

**DOI:** 10.3390/molecules25215067

**Published:** 2020-11-01

**Authors:** Freddy R. Escobar-Ledesma, Vanessa E. Sánchez-Moreno, Edwin Vera, Valerian Ciobotă, Paul Vargas Jentzsch, Lorena I. Jaramillo

**Affiliations:** 1Departamento de Ingeniería Química, Facultad de Ingeniería Química y Agroindustria, Escuela Politécnica Nacional, Ladrón de Guevara E11-253, Quito 170525, Ecuador; freddy.escobar@epn.edu.ec (F.R.E.-L.); vanessa.sanchez@epn.edu.ec (V.E.S.-M.); 2Departamento de Ciencia de los Alimentos y Biotecnología, Facultad de Ingeniería Química y Agroindustria, Escuela Politécnica Nacional, Ladrón de Guevara E11-253, Quito 170525, Ecuador; edwin.vera@epn.edu.ec; 3Rigaku Analytical Devices, Inc., 30 Upton Drive, Suite 2, Wilmington, MA 01887, USA; valerian.ciobota@rigaku.com; 4Departamento de Ciencias Nucleares, Facultad de Ingeniería Química y Agroindustria, Escuela Politécnica Nacional, Ladrón de Guevara E11-253, Quito 170525, Ecuador; paul.vargas@epn.edu.ec

**Keywords:** inulin, *Smallanthus sonchifolius*, *Agave Americana*, fructans, polysaccharide

## Abstract

Inulin is a polysaccharide of fructose widely used in the food and pharmaceutical industry due to its physicochemical properties and technological applications. Inulin from jicama (*Smallanthus sonchifolius*) and cabuya (*Agave americana*) was obtained. The steps for inulin obtention were: raw material preparation, extraction and purification. The extraction conditions were determined using a random design with three levels of stirring speed (0, 130 and 300 rpm), and a 3^2^ factorial experimental design with three levels of temperature (40, 60 and 80 °C) and solid:liquid ratio (1:2, 1:3 and 1:5 S:L). The results showed that the best extractions conditions for jicama were 130 rpm, 75 °C, 1:5 S:L and 25 min; while for cabuya were 80 °C, 300 rpm, 1:5 S-L and 100 min. The weight average molecular weight of inulin from jicama and cabuya were 5799.9 and 4659.75 g/mol, respectively. The identity of the obtained inulin from jicama and cabuya were confirmed by infrared (IR) and Raman spectroscopy. In addition, scanning electron microscopy, differential scanning calorimetry and thermogravimetry analyses were performed to characterize both inulins.

## 1. Introduction

A variety of Ecuadorian endemic plants were traditionally used due to their medicinal properties, among which jicama (*Smallanthus sonchifolius*) and cabuya (*Agave americana*) [1,2,3]. From these Andean species, some autochthonous products have been obtained, for example, “Tshawar mishki” (also known as “agua miel” or honey water), jicama syrup and cabuya syrup [4,5,6]. According to previous studies, fructans, such as inulin, and fructooligosaccharides, are the main components of these products [4,7,8,9].

Inulin is a mixture of fructose polysaccharides (fructans), composed of chains larger than 12 units, that is synthesized inside of the plant as energy reserve; short inulin chains are called fructooligosaccharides (FOS) [7]. Figure 1 shows the chemical structure of inulin, the fructose units are linked by a β (1→2) bond and the terminal molecule can be a glucose or fructose, which are called d-glucopyranosyl (GFn) and β-d-fructopyranosyl (Fm), respectively [10,11]. Different studies have shown that inulin is a fiber that acts as a bifidogenic agent, stimulating the immune system, relieving constipation and reducing the risk of colon cancer [1,10,11,12,13]. In addition, it decreases the risk of osteoporosis, prevents atherosclerosis and reduces blood glucose levels [10]. Multiple applications of inulin in food, pharmaceutical and chemical industries can be attributed to its physicochemical properties and technological applications [14,15,16,17].

Inulin is industrially obtained from various raw materials, mainly chicory roots (*Cichorium intybus*) and Jerusalem artichoke (*Helianthus tuberosus*) [10]. However, there are several studies of inulin extraction at laboratory scale using other raw materials such as burdock (*Arctium lappa*), dahlia (*Dahlia spp*), garlic (*Allium sativum*), onion (*Allium cepa*), jicama and agave tequilana (*Agave tequilana Weber var. Azul*) [18,19,20,21,22]. Jicama (also known as yacon) are composed of approximately 60 to 70% of FOS on a dry matter basis, thus being a very attractive source of FOS [7]. Although there are many studies reporting the extraction of inulin, FOS or fructans from agave species [4,23,24,25,26,27,28], only one study has reported the extraction of inulin from Tunisian Agave Americana leaves, which is focused on characterizing inulin effect on textural qualities of pectin gel [4]. Many studies on the extraction of fructans can be found in scientific literature; however, reports referring to the optimization of extraction conditions are scarce [19,29].

In this work, experimental results on the extraction, purification and characterization of inulin from Ecuadorian jicama roots and cabuya meristem are presented. The influence of inulin extraction conditions (steer speed, temperature and liquid solid ratio) on the number of soluble solids in the extracts is evaluated. The identity of the inulins from jicama and cabuya was confirmed by IR and Raman spectroscopy. Additionally, they were characterized by obtaining scanning electron micrographs and thermo-physical measurements.

## 2. Results

### 2.1. Chemical Composition of Vegetable Materials and Extraction of Soluble Matter

The chemical composition of jicama roots was: 3.7% moisture, 4.6% ash, 0.5% ether extract, 2.8% protein and 88.4% carbohydrates (estimated by difference from the other components). The estimated content of carbohydrates is in good agreement with values previously published by Choque Delgado et al. [2], who reported percentages higher than 80% of carbohydrates in jicama roots from Brazil.

Concerning the cabuya meristem, its chemical composition was: 7.7% moisture, 4.8% ash, 0.5% ether extract, 3.2% protein and 83.8% carbohydrates. On the other hand, cabuya leaves are composed of 7.6% moisture, 7.9% ash, 0.6% ether extract, 2.3% protein and 81.6% carbohydrates. Variation in the carbohydrates content between meristem and leaves was also observed in the past; Montañez-Soto et al. [25] reported higher carbohydrates contents for agave tequiliana heads (meristem) than for leaves. Both leaves and meristem were considered as sources of carbohydrates by many authors, and carbohydrates contents in these parts of the plant are variable [4,22].

In this work, since higher carbohydrates contents were observed in jicama roots and cabuya meristem, the extraction of inulin from these vegetable materials was studied. As a first stage, the extraction process was evaluated in terms of the achieved soluble matter yield. For jicama and cabuya, the achieved soluble matter yield for different extraction times are depicted in Figure 2. As shown in this figure, there is a remarkable influence of the stirring speed on the soluble matter yield for both jicama and cabuya. However, the stirring speed has a higher influence on the soluble matter yield for cabuya than for jicama. All curves have an exponential behavior that converges to an equilibrium concentration. Clearly, the stagnant fluid layer around the solid particles causes the low efficiency obtained in the absence of agitation and this effect decreases when the mixture is stirred. However, when the thickness of this layer is below a critical value, the diffusion speed becomes independent of the stirring degree, as reported in literature [30]. The equilibrium concentrations were reached after 25 and 100 min for jicama and cabuya, respectively. For purposes of this work, this time is called the “equilibrium time.” Such difference in the extraction rate can be explained considering the physical properties of the vegetable materials; jicama has a higher content of water and is less compact than cabuya (and agave species in general) [2,26].

For jicama, analysis of variance (ANOVA) and Fisher’s LSD test (Table A1 and Figure A1 of Appendix A) show no significant difference for the achieved soluble matter yields with stirring speeds of 130 and 300 rpm. On the other hand, for cabuya, the soluble matter yields in ANOVA and Fisher’s LSD test showed significant differences between all stirring levels (Table A2 and Figure A2 of Appendix A). Therefore, the selected stirring speeds were 130 and 300 rpm for jicama and cabuya, respectively.

### 2.2. Determination of Temperature and Solid:Liquid Ratio

The best conditions of temperature and solid:liquid ratio were determined by testing the fructans extraction process with the previously selected stirring speed (130 rpm for jicama and 300 rpm for cabuya) and equilibrium time (25 min for jicama and 100 min for cabuya). The response surface graphs for the extraction of fructans from jicama and cabuya are shown in Figure 3. For jicama, the beginning of an asymptotic maximum occurs at ~70 °C. This is consistent with earlier reports on the extraction of inulin from yacon (jicama) and burdock [19,21]. The contour plot located in the lower plane of Figure 3a determines that the best extraction conditions are 75 °C and a solid:liquid ratio of 1:5. For cabuya, a significant increase in soluble matter yield is observed as the temperature and the solid-liquid ratio increase, with a maximum at 80 °C and a solid:liquid ratio of 1:5. At the best extraction conditions, soluble matter yields of 16.00% (160 mg/g) and 37.25% (372.5 mg/g) were calculated (Equation (1)) for jicama and cabuya, respectively. Cabuya soluble matter yield is in good agreement with values reported previously; for example, in a previous work on fructans extraction from Agave sisalana, a total sugars content of 347.6 mg/g was reported [23].

As mentioned in Section 2.2, inulin content in jicama and cabuya extracts at best extraction conditions were calculated with Equation (2). The inulin content in the extracts was 4.538 and 19.825 g/L, and inulin yields in extracts were 2.27% (22.69 mg/g) and 9.91% (99.12 mg/g) for jicama and cabuya, respectively. These results show that cabuya had a higher yield than jicama. In previous works with similar plant material, the highest inulin yield from burdock roots was 12.31% (*w*/*w*) [21], and for yacon (jicama), an FOS (inulin) content of 9.5 g/100 mL was reported [31].

In addition, the effective diffusion coefficient was calculated as a function of temperature; Figure 4 shows the Arrhenius plots of the effective diffusion coefficients for jicama and cabuya. The two curves show a slight change of slope at 60 °C, more marked for the curve of jicama than for the curve of cabuya. These deviations were also observed in the past for beet and chicory [32] and are typically observed in extraction processes involving plant cells since there is a change in the magnitude of the resistance due to cell membranes [33].

### 2.3. Characterization of the Inulin Powder Obtained from the Fructan Extract

Scanning electron micrographs of inulin powder from jicama and cabuya are presented in Figure 5**a**–**c**. It is observed in all micrographs that inulins present amorphous zones with granular particles of spherical shape. Inulin from cabuya micrographs present some agglomerates and its structure is porous. These micrographs are similar to the obtained by Terkmane et al. [34], who stated that global artichoke inulin presents a mixture of crystallites and aggregates in its structure.

The IR and Raman spectra of the isolated inulin powders from jicama and cabuya were recorded and compared with the spectra of standard inulin (Figure 6). The band observed in the IR spectrum of standard inulin at 1635 cm^−1^ was attributed to water residues present in the sample [35]. This IR band is observed in some occasions when starch and cellulose are measured (see examples of IR and Raman spectra presented by Larkin [36]. However, the IR spectra of inulins from jicama and cabuya show other bands close to the band of water, centered at 1621, 1573 and 1540 cm^−1^, and a shoulder at 1660 cm^−1^, which could be attributed to lignin. Lignin is a complex and heterogeneous biopolymer composed primarily of p-hydroxyphenyl, guaiacyl and syringyl units [37]; however, other units could also occur [38]. Therefore, depending on the predominant units and the origin of lignin, infrared bands can occur in different ranges [39]: 1705–1720 cm^−1^ (C=O stretching modes) and ~1600–1462 cm^−1^ (Aromatic vibrations such as C=C stretching modes and CH_3_/CH_2_ bending modes). When it comes to Raman signals, the band for water at ~1635 cm^−1^ was not observed in all spectra of inulins, which is an expected aspect since this vibrational mode is weak in the Raman spectrum. The occurrence of a band at 1594 cm^−1^ in the Raman spectra of inulins from jicama and cabuya (and its absence in the Raman spectrum of standard inulin) seems to confirm the presence of lignin as impurity. Other bands of lignin (below 1462 cm^−1^) can overlap with bands of inulin and this seems to have a stronger influence in the IR spectra than in the Raman spectra of inulins from jicama and cabuya.

In the region between 1270 and 900 cm^−1^ there are bands related to C-O stretching modes [36]. The IR and Raman bands in this range are assigned to C-O stretching modes of C-OH and C-O-C from inulin. In both IR and Raman spectra of starch, cellulose and the main components of stevia extracts (stevioside and rebaudioside A) can be observed this kind of signals [36,40,41]. The IR and Raman bands observed for inulin from jicama and cabuya are similar to those obtained for standard inulin, thus being consistent to inulin. Perhaps, depending on the application of the extracted inulins, an additional step of purification should be considered to eliminate lignin impurities.

DSC and TGA thermograms of inulin from jicama and cabuya were compared with the corresponding thermograms of standard inulin (Figure 7). DSC thermograms, which are shown in Figure 6a, indicate that standard inulin starts melting around 133 °C and ends at 150 °C. Some thermal transitions can be distinguished in the thermogram of inulin from jicama; the decrease in baseline between 105 and 182 °C suggests a glass transition, and the glass transition temperature (Tg) would be around 113 °C. In an earlier study of the thermal properties of agave fructans, a glass transition event between 101.8 and 118.1 °C was identified, and a value of Tg of 109.6 °C was found [42]. In addition, the pseudo peak, with a minimum at 165 °C, could be related to a fusion process likely overlapped with devitrification. As a result, the melting process would start approximately at 113 °C and end at 182 °C, with a minimum of 165 °C and an associated melt energy of 23 J/g. In the thermogram of inulin from cabuya, a single fusion peak with an energy of approximately 44.5 J/g was observed. The melting process begins approximately at 106 °C and ends at 159 °C, with a minimum of 134 °C. The shape and width of the peak suggest a wide range of molecular weight distribution, which is in good agreement with an earlier report [43].

TGA thermograms are presented in Figure 6b. From the thermograms, a faster degradation rate was observed for standard inulin than for inulins extracted from jicama and cabuya, the inulin from cabuya being the most resistant. This could be explained based on the fact that agave fructans in cabuya have branched forms [23], and their structures are more complex to degrade than linear fructan chains. Thermograms of standard inulin show three decomposition stages, while the corresponding thermograms of inulin from jicama and cabuya show four decomposition stages. These four decomposition stages can be interpreted as follows. The first stage under 150 °C is attributed to the moisture content in the sample [42]. The mass loss for the second stage, between 150 and 240 °C, could be related to the decomposition of the biopolymer which probably starts with the breaking of the inulin chains [44]. A third long decomposition stage between 240 °C and 320 °C may be related to the decomposition of branched chains of fructans (cabuya) [23] and short inulin chains (jicama). The fourth stage, which involves a mass loss above 350 °C, could be related to the total decomposition of the main chains of inulin.

The molecular weight (Mw) of powder inulin from jicama and cabuya is shown in Table 1. The molecular weight distribution was determined by fractioning the extract by the addition of ethanol. The solubility of fructans in aqueous medium becomes unbalanced when ethanol is added to the mixture. This solubility depends also on the molecular weight of the inulin molecule. Thus, when ethanol is added to a fructan solution, they precipitate as a whitish solid. The methodology used for the fractionation describes four successive additions of alcohol (20%, 40%, 60% and 80%) to precipitate four fractions of fructans of different molecular weight. The same methodology was followed for the purified extracts from the two plants; however, for cabuya, precipitates were obtained only for three of the four fractions and for jicama only fraction four was obtained.

For jicama, the molecular weight for fraction 4 was 5799.9 g/mol with a degree of polymerization (DP) of 36. Concerning cabuya, three of the four fractions resulted in the formation of inulin powders, with DP from 23 to 61. The weight average molecular weight for inulin from cabuya was 4659.75 g/mol, while the average degree of polymerization was 29. The highest percentage of fructans (75.79%) had a molecular weight of 3718 g/mol with a DP of 23 units of anhydrous fructose. It seems that the polydispersity of crystallized inulin from jicama is narrow.

## 3. Materials and Methods

### 3.1. Vegetable Materials and Reagent

Jicama (*Smallanthus sonchifolius*) and cabuya (*Agave americana*) were used as raw materials for inulin extraction and obtained from the same cultivation zone (Guayllabamba, Pichincha, Ecuador). Jicama (roots) and cabuya (meristem and leaves) were harvested at an approximate age of 8 months and 12 years, respectively. Concerning cabuya leaves, only the first 20 cm (measured from the stem) were studied.

Ethanol, C_2_H_5_OH (LAQUIN, ≥96%), and dibasic potassium phosphate trihydrated, K_2_HPO_4_·3H_2_O (Merck, ≥99%), were used to purification process.

d-glucose (BDH, Poole, England, ≥98.5%), d-fructose (BDH, ≥98.9%), trisodium citrate dihydrate, C_6_H_5_Na_3_O_7_·2H_2_O (Merck, Darmstadt, Germany, ≥99.5%), sodium metaperiodate, NaIO_4_ (Merck, ≥99%), potassium iodide, KI (Merck, ≥99%), sodium carbonate (BDH, ≥98%) and sulfuric acid (JT Baker, Phillipsburg, USA, ≥97.99%) were used to measure inulin content in the extract by the spectrophotometric method.

Commercial Inulin (Beneo Orafti^®^ GR, with the degree of polymerization (DP) between 10 and 23 and a Mw of 1680–4100 g/mol) was used as a standard for the spectrophotometric, thermal and spectroscopic analyzes.

### 3.2. Characterization of the Vegetable Materials and Soluble Matter Extraction Conditions

Carbohydrates, proteins and ethereal extracts, as well as moisture and ash contents in jicama (roots) and cabuya (meristem and leaves), were determined by means of standard procedures [45].

The extraction stirring speed was determined for each vegetable material using a completely random design in which the soluble matter yield was the response variable. The effect of different extraction stirring speeds were compared by analysis of variance (ANOVA) using Fisher’s LSD test (P < 0.05). The soluble matter yield of extractions was determined by measuring the brix degrees using a digital refractometer (ABBE, INE-WYA-2S). The vegetable materials were gathered, washed, peeled, cut and sliced. Water extraction experiments were carried out by varying the stirring speed at three levels: 0, 130 and 300 rpm. For these experiments, the temperature and the solid-liquid ratio were kept constant at 80 °C and 1:5, respectively, which are parameters reported for similar vegetable materials in previous works [46]. The treatments were carried out in triplicate. The soluble matter yield (Y_SM_) was calculated using Equation (1):(1)$$Y_{\mathrm{SM}}\left(\%\right)=\frac{V\;\times \;C}{M_{P}}\;\times 100$$
where V is the final volume of the solution (mL), C is the soluble matter concentration after 120 min or at constant extraction (g/mL) and M_P_ is the amount of sliced vegetable material (g).

In order to determine the best conditions of temperature and solid:liquid ratio for extraction from both jicama and cabuya, response surface graphs were obtained, with the soluble matter yield as a response variable. A 3^2^ factorial experimental design was applied; three temperatures (40, 60 and 80 °C) and three solid:liquid ratios (1:2, 1:3 and 1:5) were tested. The stirring speed applied for each vegetable material was set according to results of the completely random design (see above). The statistical analyses were carried out with the software STATGRAPHICS Centurion XVI.I, version 16.1.03.

Extracts from each vegetable material were obtained by applying the best conditions of temperature and solid:liquid ratio, and then, inulin content in each extract was determined with the spectrophotometric method described by Saengkanuk et al. [47]. In this method, total fructose and free fructose are measured, and inulin content is calculated using Equation (2). Free fructose was determined by directly measuring the extract, and total fructose was determined by measuring the hydrolyzed extract. Once the inulin content was measured, inulin yield in the extract was determined with Equation (3).
(2)$$I=k\left(F_{\mathrm{tot}}-F_{f}\right)$$
where I is the inulin content (g/L), F_tot_ is total fructose content (g/L), F_f_ is free fructose (g/L) and k (0.995) is a correction factor for the glucose part of the inulin and for the water loss during hydrolysis. The absorbance measurements were performed with a UV/VIS spectrophotometer (Spectroquant^®^ Prove 100, Merck) at a 350 nm wavelength.
(3)$$Y_{\mathrm{IE}}\left(\%\right)=\frac{V\;\times \;I}{M_{P}}\times 100\;$$
where V is the final volume of the extract (L), I is the inulin content (g/L) and M_P_ is the amount of sliced vegetable material used in the extraction process (g).

Additionally, the diffusion coefficients for temperatures of 40, 60 and 80 °C and the selected conditions of stirring speed and solid-liquid ratio were determined with Equation (4) [32]:(4)$$B=\frac{C_{\left(t\right)}-C_{i}}{C_{f}-C_{i}}=1-\overset{\infty }{\underset{n=1}{\sum }}\frac{2\frac{V_{w}}{V_{s}}\left(1+\frac{V_{w}}{V_{s}}\right)}{1+\frac{V_{w}}{V_{s}}+{\left(\frac{V_{w}}{V_{s}}\right)}^{2}q_{n}{}^{2}}\;\exp\left(-\frac{{\mathrm{Dq}}_{n}{}^{2}t}{l^{2}}\right)\;$$
where B is the reduced concentration (dimensionless quantity), C_(t)_ is the soluble matter concentration (kg/m^3^), C_i_ is the initial soluble matter concentration (kg/m^3^), C_f_ is the final soluble matter concentration (kg/m^3^), V_w_ is the volume of the solvent (m^3^), V_s_ is the volume of the solid (m^3^), D is the effective diffusion coefficient (m^2^/s), t is the extraction time (s), l is the thickness of the flat wall (m) and q_n_ was determined by solving Equation (5):(5)$$\tan(q_{n})=-\frac{V_{w}}{V_{s}}\times {\;q}_{n}$$

Based on the calculated values of the effective diffusion coefficient, the Arrhenius plots were obtained with Equation (6):(6)$$D=D_{o}\;\exp\left(-\frac{E_{a}}{\mathrm{RT}}\right)$$
where D_o_ is the pre-exponential factor of the Arrhenius equation (m^2^/s), E_a_ is the activation energy (kJ/mol), R is the universal gas constant (kJ/mol K) and T is the temperature (K) [48].

### 3.3. Characterization of Inulin Powder

Inulin powders were attained from the extracts obtained as described in Section 3.2. The extracts were purified by removing proteins, lipids and other substances. Sodium phosphate was added to the extracts to reach a pH of 7.6 since this avoids hydrolysis [49], and then proteins and lipids impurities were precipitated by heating these mixtures at 70 °C for 15 min. The purified extracts were concentrated on a heating plate with constant stirring speed at 90 °C until the saturation concentration of inulin was reached (~25 °Bx at 90 °C) [49]. Once the saturation concentration was attained, the volume of the concentrated extract was measured. Ethanol was added to reach a concentration of 60% (*v*/*v*) and left to stand for 12 h at room temperature. Under these conditions, inulin precipitates, while other substances, such as saponins and phenolic compounds, are kept in solution and can be removed. The precipitates were filtered and dried at 90 °C for 4 h. Subsequently, the filtered solids (inulin powder) were weighed and characterized.

Thermogravimetry analysis (TGA) and differential scanning calorimetry (DSC) analysis were applied as a part of the characterization of inulin powder. For this purpose, a Shimadzu TGA-50 thermal analyzer was used. A mixture of inulin powder and potassium bromide was prepared to obtain a pellet for the infrared (IR) spectroscopic measurement. The pellets were pressed by 8 tons, and then, the infrared spectra were recorded using a spectrum one FT-IR spectrometer (Perkin-Elmer, Beaconsfield, United Kingdom). A spectral region of 4000–450 cm^−1^ and a resolution of 2 cm^−1^, were considered for all IR measurements. Raman measurements were performed using a handheld Raman, ProgenyTM (Rigaku Analytical Devices, Wilmington, MA, USA), spectrometer equipped with a 1064-nm Nd:YAG laser and a Peltier cooled InGaAs detector. The total laser exposure time for each Raman measurement, performed through transparent plastic bags, was 1 s with a laser power of 300 mW. The accuracy and precision of the Raman bands of the acquired spectra were below 3 cm^−1^ and 1.5 cm^−1^, respectively. The morphology of inulin was observed using a Tescan Vega LMU Scanning Electron Microscope (SEM). The obtained spectra, thermograms and micrographs were compared to those of commercial inulin (standard inulin).

The purified extracts were also used for the determination of the size of the fructans. Four fructans fractions of different average molecular weight were obtained by the addition of ethanol to both extracts; thus, the alcohol concentration ranges that correspond to each fraction are: fraction 1 (20% ethanol), fraction 2 (40% ethanol), fraction 3 (60% ethanol), fraction 4 (80% ethanol). The intrinsic viscosity of inulin (η) solutions was measured according to the ASTM D2857–16 standard (American Society for Testing And Materials, 2001). The average molecular weight (Mw) and degree of polymerization (DP) of inulin powder from the different fructans fractions, for both jicama and cabuya, were calculated using Equation (7) [34] and Equation (8) [25], respectively:(7)$$\eta =K\times M_{w}{}^{a}$$
where K and a are the empirical constants of Mark-Hounwink, 6.76 × 10^−3^ and 0.71, respectively.
(8)$$M_{wINU}=162\times DP_{INU}$$
where M_WINU_ is the average molecular weight of inulin and 162 is the molecular weight of a fructose or glucose residue that makes up the inulin molecule.

Finally, the weight average molecular weight was calculated according to Equation (9) [50]:(9)$$W_{Mw}=\overset{n}{\underset{i=1}{\sum }}w_{i}\times M_{wi}$$
where W_Mw_ is the weight average molecular weight and w_i_ is the weight fraction of different fructan fractions having an average molecular weight of M_Wi_.

## 4. Conclusions

The content of carbohydrates was lower in cabuya leaves than in cabuya meristem. Therefore, cabuya meristem was used for the extraction of fructans.

Both temperature and solid:liquid ratio influence the extraction of fructans. The soluble matter yield increased at a high temperature and solid-liquid ratio; for cabuya meristem, the maximum soluble matter yield was achieved at 80 °C and a solid:liquid ratio of 1:5, while for jicama roots the best conditions were 75 °C and a solid:liquid ratio of 1:5.

SEM micrographs show that both jicama and cabuya inulins present amorphous zones with granular particles of spherical shape.

The identity of inulin powder from jicama and cabuya was confirmed by IR and Raman spectroscopy. Characteristic bands of inulin were observed in both IR and Raman spectra of these materials; however, some bands attributable to lignin were also observed. It seems that lignin impurities persist in the final inulin powder. As an option, the use of methanol instead of ethanol could improve the purification of inulin. Moreover, the application of alternative methods of extraction such as ultrasound-assisted extraction, microwave extraction and pulsed electric field may minimize lignin residues. These are some strategies that future works should focus on to avoid these kinds of impurities in the final product.

The identity of inulin powders was also confirmed by means of the DSC and TGA thermograms. Melting points of inulin from jicama and cabuya showed similar values to the standard inulin. TGA thermograms of inulin from jicama and cabuya showed four decomposition stages, while for standard inulin only three decomposition stages were observed. Such a difference could be related to the characteristics of inulin chains.

It was found that the polydispersity of crystallized inulin from jicama is narrower than from cabuya. For inulin from jicama, the weight average molecular weight was 5799.9 g/mol and the degree of polymerization in mass was 36. For inulin from cabuya, the weight average molecular weight was 4659.75 g/mol and the average degree of polymerization in mass was 29; the highest percentage of fructans (75.79%) had a molecular weight of 3718 g/mol and a DP of 23 units of anhydrous fructose.

## Figures and Tables

**Figure 1 molecules-25-05067-f001:**
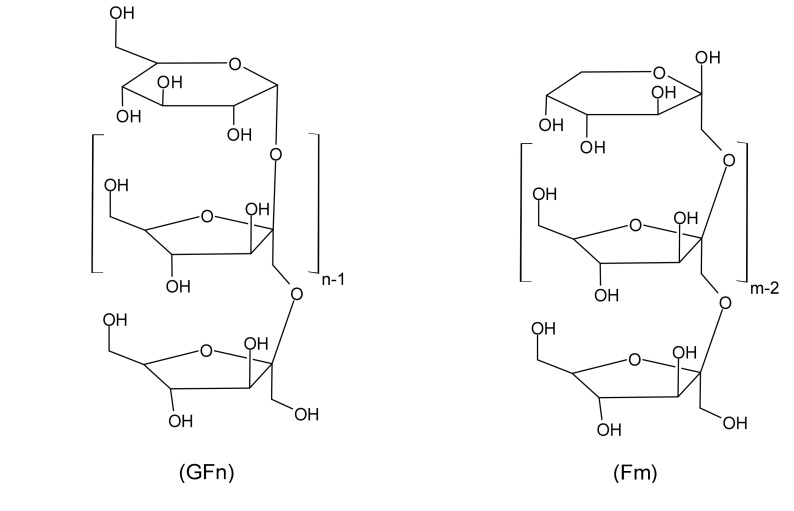
Structure of Inulin.

**Figure 2 molecules-25-05067-f002:**
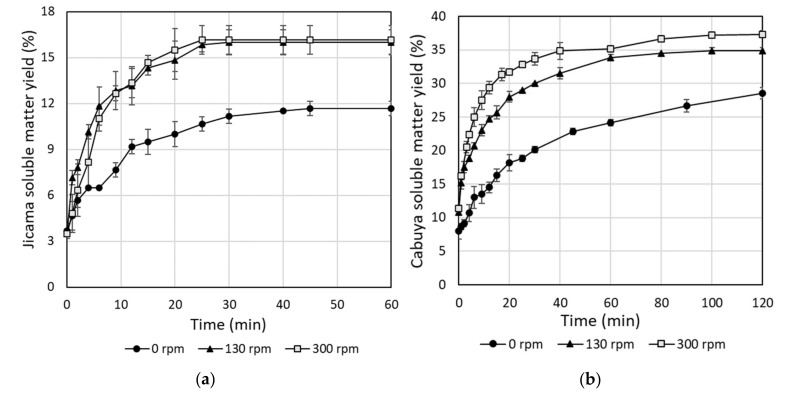
Variation of soluble matter yield as function of extraction time for: (**a**) Jicama roots; and (**b**) Cabuya meristem. Extraction conditions: temperature = 80 °C; solid:liquid ratio = 1:5.

**Figure 3 molecules-25-05067-f003:**
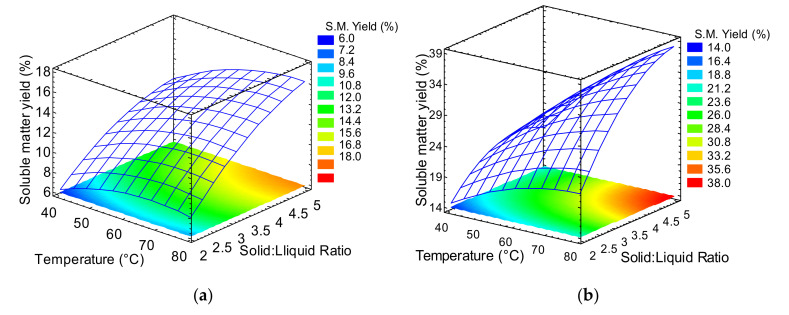
Response surface graphs of soluble matter yield as function of temperature and solid:liquid ratio (2 = 1:2, 3 = 1:3 and 5 = 1:5): (**a**) jicama (130 rpm), and (**b**) cabuya (300 rpm).

**Figure 4 molecules-25-05067-f004:**
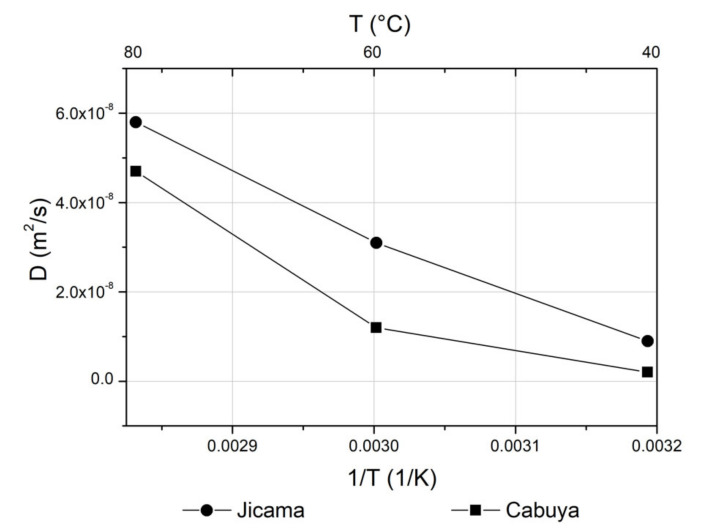
Arrhenius plots of the effective diffusion coefficients for jicama and cabuya.

**Figure 5 molecules-25-05067-f005:**
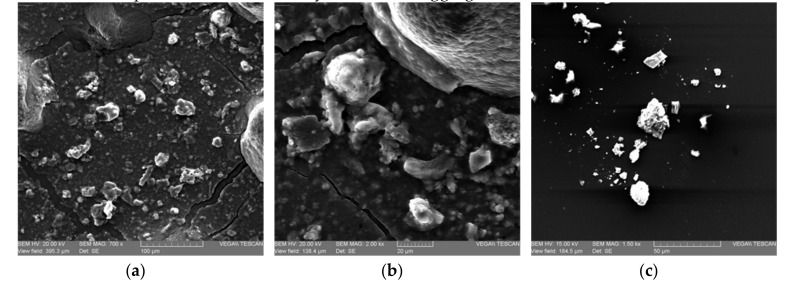
Scanning electron micrographs of inulin powder from: jicama at a magnification factor of: (**a**) 700× and (**b**) 2000×; and cabuya at a magnification factor of (**c**) 4000×.

**Figure 6 molecules-25-05067-f006:**
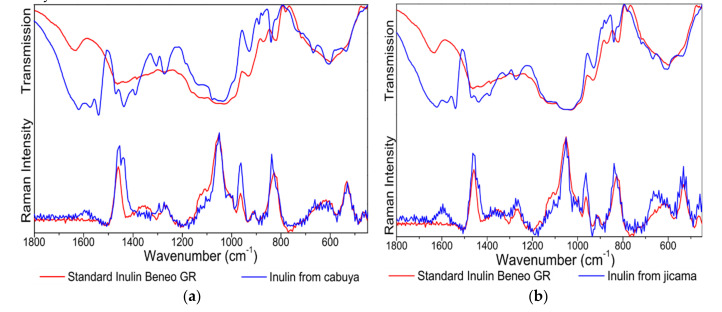
Comparison of the IR and Raman spectra of standard inulin and inulin from: (**a**) jicama and (**b**) cabuya.

**Figure 7 molecules-25-05067-f007:**
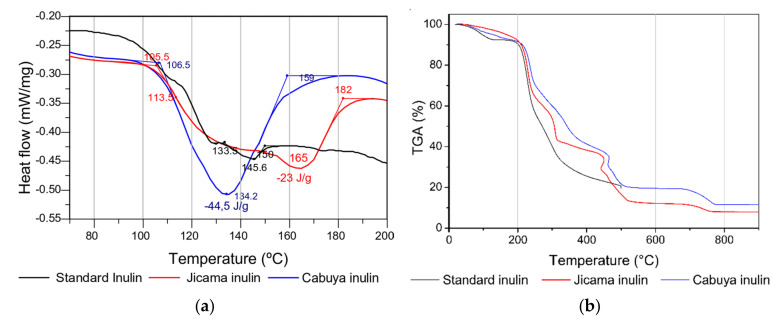
Plots of the thermal analyses of the commercial inulin (standard) and extracted inulin: (**a**) DSC and (**b**) TGA.

**Table 1 molecules-25-05067-t001:** Molecular weight of inulin powder obtained from jicama and cabuya inulin powders.

Plant	Fraction (% EtOH/H_2_O *v*/*v*)	M_w_ (g/mol)	DP
Jicama	Fraction 4 (60–80) ^1^	5799.9 ± 358.8	36
Cabuya ^2^	Fraction 1 (0–20)	9489.6 ± 684.5	59
	Fraction 2 (20–40)	6144.1 ± 237.6	38
	Fraction 3 (40–60)	3718.0 ± 486.4	23

^1^ Inulin powder from jicama was obtained by the addition of ethanol like cabuya; however, no crystallization was observed for fractions 1 to 3. Crystallization was observed only for fraction 4. ^2^ No crystallization was observed for fraction 4 of inulin from cabuya.

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
