# Peer review of "Extraction of Inulin from Andean Plants: An Approach to Non-Traditional Crops of Ecuador"

_molecules, 2020, doi:10.3390/molecules25215067_

Round 1

Reviewer 1 Report

In my opinionthis manusript is interesting, however needs major revision before publication.

Specific coments:

  1. What is the statistical analysis used?- you did dot mentioned about it in methodology, but in results you write e.g. " showed significant differences between all stirring levels". Only making statistically measurements we can say that the difference is signifficant or not. Also Please add standard deviations to the results (in tables and figures)
  2. Table 1 - You mentioned faction 4 in the descriptions below the table - but in the table there are only 3 factions for Cabuya and one for jicama. They are marked with 1-3 - is something missing from the table? Please check it.

Author Response

Subject: Submission of the revised article for Molecules

Dear Ladies and Gentlemen,

Please find enclosed the revised manuscript for an article in Molecules:

“Extraction of inulin from Andean plants: an approach to non-traditional crops of Ecuador” by Freddy R. Escobar-Ledesma, Vanessa E. Sánchez-Moreno, Edwin Vera, Valerian Ciobotă, Paul Vargas Jentzsch and Lorena I. Jaramillo.

First of all, we like to thank the reviewers for the helpful comments.

We carefully considered all comments and changed the corresponding paragraphs in the text and answered the reviewer comments. Please find attached the answers to the reviewer’s comments.

Yours Sincerely,

Lorena I. Jaramillo.

Reviewer 2 Report

The paper from Escobar-Ledesma focuses on the optimization of the extraction procedure of the natural component inulin from plants from Ecuador (jicama and cabuya). The article is in general very well written and the topic is surely interesting. Several analytical/spectroscopical techniques were adopted by the authors and the results are well presented. I report some comments below.

Introduction

Introduction is very well written and clear. Although, since Molecules is a very chemistry-oriented journal, I would suggest at least adding the chemical structures of fructose/inulin.

Results

Are content percentage (lines 71-73) fully referenced? Please check. Figure 2 is surely interesting and well represents the results. Nevertheless, resolution is not optimal and grey lines appear at the bottom of the two panels. Please check. I am concerned about the serious differences in the DSC plots with respect to the standard. The authors commented on this point but I would like to check if the same behavior was observed in multiple samples of the extract. Is the measurement reproducible?

Conclusion

This section well resumes the findings of the paper. The authors should comment more precisely on the presence of lignin impurities, why they think that these impurities are present and how they could suggest to further purify the extracts for potential future uses.

Author Response

(The authors gave the same response as above.)
